# Conditions Necessary for the Transfer of Antimicrobial Resistance in Poultry Litter

**DOI:** 10.3390/antibiotics12061006

**Published:** 2023-06-03

**Authors:** Aaron Oxendine, Allison A. Walsh, Tamesha Young, Brandan Dixon, Alexa Hoke, Eda Erdogan Rogers, Margie D. Lee, John J. Maurer

**Affiliations:** 1School of Animal Sciences, Virginia Polytechnic Institute and State University, Blacksburg, VA 24060, USA; acoxend@vt.edu (A.O.); raquel.young11@gmail.com (T.Y.); brandandxn@gmail.com (B.D.); alexa.hoke@gmail.com (A.H.); 2Department of Biomedical Science and Pathobiology, Virginia Polytechnic Institute and State University, Blacksburg, VA 24060, USA; eda_rogers@lifenethealth.org (E.E.R.); mlee2@vt.edu (M.D.L.)

**Keywords:** plasmids, litter, conjugation, *Salmonella*

## Abstract

Animal manures contain a large and diverse reservoir of antimicrobial resistance (AMR) genes that could potentially spillover into the general population through transfer of AMR to antibiotic-susceptible pathogens. The ability of poultry litter microbiota to transmit AMR was examined in this study. Abundance of phenotypic AMR was assessed for litter microbiota to the antibiotics: ampicillin (Ap; 25 μg/mL), chloramphenicol (Cm; 25 μg/mL), streptomycin (Sm; 100 μg/mL), and tetracycline (Tc; 25 μg/mL). qPCR was used to estimate gene load of streptomycin-resistance and sulfonamide-resistance genes *aadA1* and *sul1*, respectively, in the poultry litter community. AMR gene load was determined relative to total bacterial abundance using 16S rRNA qPCR. Poultry litter contained 10^8^ CFU/g, with Gram-negative enterics representing a minor population (<10^4^ CFU/g). There was high abundance of resistance to Sm (10^6^ to 10^7^ CFU/g) and Tc (10^6^ to 10^7^ CFU/g) and a sizeable antimicrobial-resistance gene load in regards to gene copies per bacterial genome (*aadA1*: 0.0001–0.0060 and *sul1*: 0.0355–0.2455). While plasmid transfer was observed from *Escherichia coli* R100, as an F-plasmid donor control, to the *Salmonella* recipient in vitro, no AMR *Salmonella* were detected in a poultry litter microcosm with the inclusion of *E. coli* R100. Confirmatory experiments showed that isolated poultry litter bacteria were not interfering with plasmid transfer in filter matings. As no R100 transfer was observed at 25 °C, conjugative plasmid pRSA was chosen for its high plasmid transfer frequency (10^−4^ to 10^−5^) at 25 °C. While *E. coli* strain background influenced the persistence of pRSA in poultry litter, no plasmid transfer to *Salmonella* was ever observed. Although poultry litter microbiota contains a significant AMR gene load, potential to transmit resistance is low under conditions commonly used to assess plasmid conjugation.

## 1. Introduction

The United States produces ~20 million tons of poultry manure each year [1,2]. Birds are raised on wood shavings and other plant products as bedding, on which the animals defecate. With time, this bedding material is broken down by microbial activity. The material, referred to as poultry litter, is a major by-product generated in poultry meat production and is highly valued as fertilizer. Animal manures, including poultry litter, are often used as soil amendments on organic produce farms [3]. However, application of animal manures has public health risks, as they harbor zoonotic pathogens [4,5,6]. Over the past 40 years, there has been a significant increase in foodborne outbreaks associated with the consumption of produce [7,8,9] and there have been several outbreaks tied directly to the application of animal manures to fields [10,11,12].

Animal manures also contain a diverse and abundant antimicrobial resistome [13]. Poultry litter is a complex microbial community consisting of more than just fecal bacteria, and consists of member species belonging to the *Lactobacillaceae*, *Aerococcacea*, *Carnobacteriaceae*, *Staphylococcaceae*, *Corynebacteriaceae*, and *Microcinaceae* [14]. The Gram-negative proteobacter that are food safety threats, including *Escherichia coli*, *Campylobacter*, and *Salmonella*, are a minor component of the litter microbiota. Poultry *Escherichia coli* generally exhibit multi-drug resistance (MDR) [15] and most isolates possess integrons, mobile genetic elements (MGEs) that capture antimicrobial-resistance gene cassettes [16]. Gram-negative enterics isolated from poultry generally possess integrons [17]. There is a sizable antimicrobial resistome in poultry litter [18,19], including integrons and their associated resistance genes [18]. However, integrons are present in a diverse group of poultry litter bacteria, especially the abundant Gram-positives. The MGE integron’s integrase *intI1* and commonly associated streptomycin-resistance gene *aadA1* is identical in diverse bacterial species such as *Corynebacterium*, *Staphylococcus*, *E. coli*, and *Salmonella enterica*. This broad distribution of specific MGEs, and associated antimicrobial resistance (AMR) genes, within the litter microbiota suggests conjugative MGE at play in their dissemination. There is reasonable concern over the spillover of these resistances into soil amended with poultry litter and potential spread to bacteria that inhabit the soil consortium. Most concerning is the potential acquisition of antimicrobial resistances by pathogens in this environment. 

Antimicrobial resistant bacteria and fungi cause 2.8 million infections and 35,900 deaths annually in the United States [20]. Drug-resistant *Campylobacter* and non-typhi *Salmonella* alone account for twenty-three percent of these infections [20]. While gastroenteritis associated with both pathogens is self-limiting in adults, children (<5 years old) and the elderly (>65 years old) are most likely to require some medical intervention [21,22,23]. For salmonellosis, antibiotic therapeutics can lessen the severity and duration of illness [23]. What is therefore troubling is not just the increase in resistance to antibiotics generally used to treat gastroenteritis [20], but that the reported treatment failures are tied to resistance [24,25,26,27,28]. While *Campylobacter* resistance to some prescribed antimicrobials is due to spontaneous mutations in the antibiotic’s target [29,30], antimicrobial resistance in non-typhi *Salmonella* depends on mobile, conjugative plasmids [27,31]. Multi-drug resistance among *Salmonella* isolated from the poultry farm environment often depends on mobile conjugative plasmids [32,33,34,35,36]; therefore, transfer of resistance requires physical, cell-to-cell contact [37]. In a few instances, these plasmids are related to those present in unrelated microbes [38,39,40] and, furthermore, *Salmonella* and Gram-positive bacteria in poultry litter have been shown to possess the same integron-associated antimicrobial-resistance genes [18,41]. *Salmonella* can also acquire antimicrobial resistance while colonizing chickens, even in the absence of antibiotic selection pressure [42]. Since poultry litter has a sizable and diverse resistome, resistance can therefore potentially be transmitted to pathogens that encounter this reservoir, as there is a significant likelihood of recipient cells contacting a plasmid-donor cell. This study assessed the potential for transmission of antimicrobial resistance from the litter resistome to a susceptible *Salmonella* strain and the parameters associated with plasmid transfer in poultry litter.

## 2. Results

### 2.1. A Sizable Litter Resistome Did Not Result in Transfer of Antimicrobial Resistance from Poultry Litter to Salmonella 

Class 1 integrons and associated antimicrobial-resistance genes *aadA1* and *sul1* are abundant in poultry litter [18]. The integron-associated sulfonamide-resistance gene, *sul1*, was abundant in the poultry litter used in these experiments. Overall, 35 to 7% of the bacterial population possessed *sul1* with the integron-associated, streptomycin-resistance gene *aadA1* present in just 0.43% and 0.04% of the population, for experimental vs. commercial poultry litter, respectively (Table 1) (*p* < 0.001). Of the total poultry litter population, ~0.5 to 4% of litter bacteria from experimental vs. commercial sources, respectively, were resistant to streptomycin (*p* = 0.0604). There was also a high level of tetracycline resistance in poultry litter, with 5% of the total bacterial population exhibiting resistance (*p* = 0.3196). However, resistance to chloramphenicol was 4 Log_10_ less than tetracycline resistance in bacteria from the experimental litter and <2.99 Log_10_ for commercial sources. Similarly, ampicillin resistance also varied between the two litter sources. The poultry litter sources also differed in the level of Gram-negative enterics (<3 vs. 4 Log_10_ CFU/g). Because of between-group (farms) and within-group variability, as determined by ANOVA, litter samples were subsequently pooled to assess transferability of AMR from litter to a recipient *Salmonella* strain. With the high level of resistance genes and phenotypes, we expected facile transfer of antimicrobial resistance (AMR) from poultry litter bacteria to a susceptible bacterial strain. However, we did not detect transfer of antimicrobial resistance between the poultry litter microbiota and an antimicrobial-sensitive *Salmonella* Typhimurium strain, regardless of the abundance of Gram-negative enterics present in the litter community in filter matings (Table 2). We hypothesized that the AMR transfer may occur in specific conditions optimal for conjugative plasmids present in the litter community. Therefore, we tested multiple conditions that varied media strength, growth temperature, and length of contact time in filter matings in an attempt to identify conditions enabling AMR transfer from litter bacteria to *Salmonella*.

### 2.2. What Conditions Are Optimal for Transfer of AMR from Poultry Litter Resistome to Salmonella?

*Escherichia coli* containing conjugative plasmid R100-1 served as a plasmid donor control in experiments to optimize AMR transfer to the *Salmonella* Typhimurium LT2 pSLT^−^ recipient. In filter matings, the earliest detection of R100 plasmid transfer to *Salmonella* was six hours at 37 °C (Figure 1). Peak plasmid transfer was observed for overnight incubation at 37 °C, where the conjugation frequency was 0.48. Therefore, all subsequent filter conjugation experiments were performed with an incubation of at least 24 h. Growth temperature (25 °C vs. 37 °C) and length of filter incubation (24 or 72 h) were then varied to identify parameters that may result in the highest poultry litter AMR transfer rate using *E. coli* with R100 as a donor control (Table 3; Figure 2). AMR transfer was observed using the *E. coli* plasmid-donor control in filter matings at 37 °C for a minimum of 24 h. No plasmid transfer of the R100 donor control was observed at 25 °C incubation for up to 72 h (Figure 2). Another potential factor in plasmid transfer is media composition and richness where bacterial growth rate influences gene transfer [44,45,46]. To address this possibility, tryptic soy broth and agar were diluted 10-, 100-, and 1000-fold and used in filter matings at 37 °C (Figure 3). Transconjugant abundance correlated with media strength (Figure 3A) and ultimately recipient abundance (Figure 3B). Dilute media (100- to 1000-fold) only reduced conjugation frequency of *E. coli* R100 by ~10-fold. Therefore, subsequent experiments used a variety of conditions to detect litter AMR transfer to *Salmonella*.

No transfer of plasmids from poultry litter bacteria was observed in these conditions. However, plasmid transfer was observed when the *E. coli* R100 plasmid donor was included with the poultry litter bacteria in filter matings at 37 °C (Table 2). *Salmonella* abundance and R100 conjugation frequency were reduced ~2 log_10_ using commercial poultry litter compared to litter bacteria from the research flock (*p* < 0.01). No ampicillin resistance was observed in transconjugants, despite ~5% of the commercial litter bacteria being resistant; furthermore, only resistance phenotypes linked to R100 were observed, suggesting that no other plasmids were transferred to *Salmonella*. 

### 2.3. Poultry Litter Bacteria Are Not Inhibiting Plasmid Transfer between Escherichia coli and Salmonella

We hypothesized that failure to observe AMR transfer from litter bacteria to *Salmonella* might be due to interference. Conjugation requires physical cell-to-cell contact for plasmid transfer to occur. *Salmonella* is often present at low levels in poultry litter [47]. If litter bacteria that possess conjugative AMR plasmids and antibiotic-susceptible recipient cells are both minor populations, the probability of transfer would be low. In order to investigate the role of donor abundance, conjugation frequency was examined using donor *E. coli* containing R100, starting at 10^6^ cell density and diluting 10-fold to extinction. Filter matings were performed with high (10^8^) or low (10^6^) levels of poultry litter bacteria relative to recipient concentrations (10^6^ vs. 10^8^, respectively) with decreasing abundance of the *E. coli* donor (Figure 4). R100 plasmid transfer (conjugation frequency: 6.06 × 10^−6^) to *Salmonella* was observed even at levels where the donor and recipient strains were at their lowest starting cell density (10:10^6^) relative to the total poultry litter bacteria (10^8^). Transconjugant abundance positively correlated (R^2^ = 0.9632) with the initial starting cell density of the plasmid donor. Therefore, while AMR gene abundances are high in poultry litter, these results suggest that the resistance genes may not reside on conjugative, mobile genetic elements capable of transfer to, persistent in, or expressed in *Salmonella*.

### 2.4. The Contribution of Donor Plasmid and Donor Strain Type to Antibiotic Resistance Transmission in Poultry Litter

The poultry house environment is generally maintained at 75 °F (25 °C) when rearing chickens. Poultry house temperatures are slightly elevated to 30 °C during brooding when hatchlings are first placed in the flock house [48]. In an effort to better mimic the flock house environment, gene transfer studies were performed at 25 °C. However, no R100 plasmid transfer to *Salmonella* was observed in the poultry litter at 25 °C, despite the persistence of both the donor *E. coli* and recipient *Salmonella* in the poultry litter microcosm for 7 days (Figure 5). While there were between-group differences (donor alone vs. donor and recipient) in *E. coli* counts on nalidixic acid alone (strain or strain with R100) or with chloramphenicol (donor with plasmid) (ANOVA *p* < 0.01), no in-group differences were observed. Failure to observe transfer of pR100 was, therefore, not due to its loss but rather related to temperature, as transfer readily occurred at 37 °C, and not 20 °C, in vitro. Therefore, several AMR conjugative plasmids, belonging to different plasmid incompatibility groups, were examined for their ability to transfer resistance at 25 °C. IncI and IncW plasmids were capable of transferring resistance to *Salmonella* at 25 °C with transfer frequencies of 10^−4^ to 10^−9^ (Table 4). Conjugative plasmid pRSA was chosen for further study due to derepression of the *tra* operon and broad-host range [49,50]. Because the *E. coli* donor strain MC4100 contains several mutations including *recA1*, *thi01*, and *relA1* that may place it at a metabolic disadvantage in poultry litter, we substituted it for *E. coli* 1932, a wild-type prototroph. *Escherichia coli* 1932 containing pRSA was mixed with *Salmonella* in a poultry litter microcosm at 25 °C. However, the *E. coli* 1932 strain did not persist in the litter past 3 days, while the *Salmonella* recipient strain persisted up to 7 days (Figure 6A). No transconjugants were observed using donor strain 1932 and plasmid pRSA in poultry litter. *E. coli* 1932 is a human isolate and apparently less able to adapt to the litter environment than poultry isolates. Plasmid pRSA was then moved into the nalidixic-resistant chicken *E. coli* isolate, 5651 [51]. Repeating the previous experiment, *E. coli* 5651 containing pRSA was used as the plasmid donor. This *E. coli* strain persisted longer than the human isolate; however, it was not detected on day 7 (Figure 6B). There were no significant differences in *E. coli* counts on media containing nalidixic acid or kanamycin (Student’s *t*-test *p* > 0.05). No transconjugants were observed over the 7-day mating period in the poultry litter microcosm.

## 3. Discussion

There was a sizable poultry litter resistome, especially with regard to phenotypic resistances to streptomycin and tetracycline and a corresponding abundance of *aadA1* (streptomycin resistance) and *sul1* (sulfonamide resistance). Poultry litter has been reported to contain an abundant and diverse resistome [18,53,54]. Many AMR genes are also linked to mobile genetic elements (MGE) [53,55]; however, integrons and some transposons are not self-transmissible and depend on conjugative plasmids, phages or natural transformation for dissemination. In this study, we did not detect transfer of antimicrobial resistance between the poultry litter bacteria and an antimicrobial-sensitive *S.* Typhimurium recipient strain, despite the abundance of Gram-negative enterics present in the litter community. Numerous AMR genes and associated MGE are shared among disparate bacterial members of the litter microbiota, including *Salmonella* [18]. Comparative genome analyses of various multi-drug-resistance (MDR) plasmids bare out a common origin amongst different bacterial species harboring the same or similar conjugative MDR plasmids [38,56,57].

Because we did not detect AMR gene transfer to *Salmonella*, we investigated the role of donor strain, plasmid type, and environmental conditions on transfer rates and donor strain persistence. The physiological state, of donor and recipient, is an important parameter to plasmid transfer [44,45,46]. Shafieifini et al. reported higher conjugation frequency with lower strength Mueller–Hinton broth, indicating that slower growth rates may enhance transfer [45]. *Salmonella* virulence plasmid belonging to the incompatibility group IncFII was shown to be transferable only when the plasmid donor was grown in a minimal medium [58]. Fernandez-Astorga et al. reported a decline in transconjugants and conjugation frequency with diluted media [59]. These differences may reflect plasmid or donor/recipient strain type. However, varying temperature and media concentration did not result in increased AMR transfer.

Some conjugative plasmids respond to mate-sensing signal peptides produced by potential recipients. In this system, the donor also produces an antagonist to the signal, inhibiting plasmid transfer at high donor-to-recipient ratios [60]. At low cell densities, plasmid transfer occurs at a high rate until the donor and transconjugant population density produces enough antagonists to inhibit conjugation. Others have also found that the quorum-sensing autoinducer acyl homoserine lactone can increase plasmid transfer in a dose-dependent manner [61]. Autoinducers can affect bacterial motility [62,63]. Autoinducer AI-2 can even act as a chemoattractant [62] and induce biofilm formation [64], where plasmid transfer can occur [46]. Quorum-sensing has also been shown to contribute to plasmid transfer in microbial communities [65]. For these reasons, we also varied *E. coli* and litter bacteria donor concentrations in filter matings and used the litter microcosm itself in order to provide signaling molecules for the matings. Although these conditions did not result in plasmid transfer in our experiments, plasmid transfer to Gram-negative enterics has been well documented in vivo [42,66,67,68,69,70,71] and ex vivo [72], including studies demonstrating acquisition of resistance from the resident microbiota [42,67,72]. The inability to document plasmid transfer from the poultry litter microbiome to *Salmonella* may be attributed to (1) the absence of AMR conjugative plasmids in the bacterial population [73]; (2) limited transfer potential of conjugative plasmids or plasmid replication outside their donor host, kin, or evolutionarily related bacteria [37,68,74,75,76]; or (4) genetic barriers to plasmid acquisition [77], including exclusion by resident plasmids [78]. However, we did not detect plasmid transfer in the poultry litter microcosm even when an *E. coli* plasmid donor was included. In order to determine if failure of plasmid transfer was due to plasmid type, we utilized a number of conjugative plasmids of different incompatibility groups. IncF plasmid R100 transfer is optimal at 37 °C, and no transfer was ever detected at room temperature. IncI plasmids are commonly found in *Salmonella* from non-food animal sources and contain the full conjugation machinery for plasmid transmission [78]. The IncW plasmid pRSA was chosen based on its use in comparable studies [49,79] and its ability to transfer at 25 °C. However, transfer was not observed when *E. coli* with conjugative plasmid pRSA was used in poultry litter. In this case, *E. coli* strain type became a confounding factor, as *E. coli* 1932 died off after 3 days. While switching to a poultry isolate, *E. coli* 1932, resulted in improved persistence, no plasmid transfer was detected. Others have also reported the influence of donor strain and plasmid type on AMR transfer [68,69,74,80,81]. While the focus here is on donor and plasmid, recipient background can significantly impact plasmid acquisition [74,78]; therefore, we selected *Salmonella* LT2 pST^−^ which has been cured of the resident F plasmid that can act as barrier to plasmid transfer.

Despite our earlier findings of a low rate of plasmid transfer to *Salmonella* in vivo [42], the rate of acquisition of AMR from the litter microbiota was too low to detect in these experiments. This is probably most likely because the poultry litter used in this study did not support or inhibit *Salmonella* or *E. coli* growth. Alternatively, the poultry litter microbiota may produce factors that interfere with plasmid abundance or transfer [73]. Bacterial growth is central to AMR transfer [45], as energy is needed for plasmid transmission as well as DNA synthesis machinery needed to produce the complementary strand of the transferred plasmid DNA [82]. Because no increase in donor or recipient cell density was ever observed in poultry litter even at low cell densities, transfer may be a rare event. However, even in an *E. coli* strain that persisted in poultry litter, the plasmid was lost after 3 days of incubation in the microcosm. We previously reported similar findings for *S*. Newport AMR plasmid in chickens administered *E. coli* donors with antibiotic selection pressure [42]. Others have reported the impact of donor strain background on plasmid transfer [69,74,80,81]. Plasmid transfer has been previously reported in poultry bedding material where conjugation frequencies varied depending on bedding or presence of inhibitory chemical residues [83,84]. Guan et al. reported plasmid transfer but with a higher rate in chicken manure than compost where donor and recipient cell abundance was affected by compost temperature [85]. Tecon et al. reported that the frequency of cell–cell contact that led to plasmid transfer was a function of water activity; the drier the cell matrix, the more likely plasmid transfer occurs [86]. Therefore, the physical nature of the litter matrix may also impact on plasmid transfer indicating that litter management may have a large impact of the likelihood of resistance transfer to foodborne pathogens such as *Salmonella*. Plasmid transfer in litter requires (1) permissive microbiota, bacteria strain, and plasmid types; (2) bacteria growth; (3) water activity that favors biofilm formation; and antibiotic selection pressure that favors emergent AMR in pathogens inhabiting this environment. Future studies should reveal whether certain litter types or management practices such as top dressing, litter amendments, or deep litter systems are more likely to promote AMR dissemination.

## 4. Materials and Methods

### 4.1. Extraction of Bacteria from Litter

Bacteria were extracted from litter and separated from detritus using spin columns as previously described [87]. For each sample, one gram of litter was weighed out and placed in a 50 mL conical tube with 10 mL of 0.1% Tween 80 in sodium phosphate buffer (3.4 mM NaH_2_PO_4_, 46.6 mM Na_2_HPO_4_; pH 7.4). The tube was then placed in the arm of a Fisherbrand™ “wrist-action” Flask Shaker (Fisher; Waltham, MA, USA) and processed at maximum speed for five minutes. Spin columns were constructed by removing the plunger and flaps from a 20 mL syringe then placing it within a 50 mL conical tube. Sterile 4 × 4-inch gauze pads were placed into the 20 mL syringe to form the column matrix. The columns were pre-wet with 10 mL 0.1% Tween 80 in 50 mM sodium phosphate buffer (pH 7.4). The litter samples were then poured into the spin columns which were centrifuged at 700× *g* for 1 min. The spin column filtrate was centrifuged a second time at 2000× *g* for 30 min at 4 °C to pellet the bacteria. The supernatant was discarded and the pellet was resuspended in 1 mL freezer stock medium (15% glycerol, 1% peptone). The sample was equally split into two sterile 1.5 mL capacity microfuge tubes and cells were pelleted by centrifugation at 7500× *g* for 15 min. The supernatant was discarded from each tube. One pellet was stored at −20 °C until DNA extractions. The second pellet was resuspended in 750 μL freezer stock media and transferred to 1.5 mL cryovials stored at −80 °C.

Poultry litter used in this study was obtained from two sources: experimental broiler flocks and commercial broiler chicken houses in the southeastern United States. Litter obtained from the experimental flock was collected following the 3rd successive chicken flock raised on built-up litter. Random grabs of poultry litter were collected from vacated experimental pens and pooled. Poultry litter was also obtained from four commercial broiler chicken farms submitted to us by a third party. The chicken breeds were typical Ross or Ross/Cobb hybrids used for the production of meat birds (broilers). Bedding for experimental and commercial flocks was pine shavings. Commercial broiler production in the US typically uses a built-up litter system: windrowing to compost litter before its spread, to which it is top dressed with fresh pine shavings before the placement of the next flock.

### 4.2. Enumeration of Poultry Litter Bacteria: Total Aerobic Counts, Gram-Negative Enterics, and Antimicrobial-Resistant Bacterial Count

Litter bacterial counts were performed by diluting cell suspensions 10-fold in buffered saline gelatin (BSG) [88] with a dilution range of 10^−1^ to 10^−6^ and 10 μL was spotted, in triplicate, onto an agar plate surface. Total aerobic counts were acquired by plating dilutions on tryptic soy agar (TSA) (Sigma Aldrich; St. Louis, MO, USA); Gram-negative enteric counts were determined by plating dilutions on MacConkey agar (Fisher). Antimicrobial resistant bacterial counts were determined by plating dilutions on TSA plates with ampicillin (25 μg/mL), chloramphenicol (25 μg/mL), kanamycin (25 μg/mL), streptomycin (100 μg/mL), and tetracycline (10 μg/mL). Plates were incubated overnight at 37 °C. Counts were enumerated and recorded for each dilution.

### 4.3. In Vitro Conjugation

Bacterial strains and plasmids used in this study are described in Table 5. *Escherichia coli* M4100 containing incFII conjugative plasmid pR100-1 or poultry litter bacteria served as donors in the initial conjugation experiments. The *Salmonella enterica* Typhimurium strain pSLT^−^ was chosen as recipient based on high plasmid-transfer frequencies reported for this strain in filter matings with *E. coli* donors bearing IncFI or IncFII plasmids [78]. *Salmonella* recipient and *E. coli* donor strains were grown as standing, overnight cultures at 37 °C in 5 mL tryptic soy broth (TSB; Sigma Aldrich) supplemented with 10 mM MgSO_4_. The mating mix consisted of *E. coli* MC4100 with R100-1 (5 μL, ~10^6^ CFU) or poultry litter bacteria (5 μL; ~10^6^ CFU) with *S*. Typhimurium pSLT^−^ recipient strain (50 μL; ~10^7^ CFU) in 5 mL of 10 mM MgSO_4_. Cells were collected on a 0.45 μm pore size cellulose filter membrane (Millipore Sigma; Burlington, MA, USA), which was aseptically placed, cell side up, on M9 agar containing 0.2% glucose [78]. Pure cultures with donor or recipient strain alone were similarly treated and included as controls. After overnight incubation at 37°C, a cell suspension was made by vortexing the filter in 5 mL of 10 mM MgSO_4_. The cell suspension was diluted 10-fold in buffered saline gelatin (BSG) [88] and plated on TSA plates containing the appropriate antibiotic for enumerating *Salmonella* alone (rifampicin, 64 μg/mL) or *Salmonella* transconjugants from matings with *E. coli* pR100-1 (rifampicin, 64 μg/mL; chloramphenicol, 25 μg/mL) or litter bacteria (rifampicin, 64 μg/mL; ampicillin, chloramphenicol, streptomycin, or tetracycline at concentrations stated in Section 4.2). The conjugation frequency was determined from the number of transconjugants divided by recipients, averaging the results of triplicate matings [78].

The impact of temperature (25 °C vs. 37 °C) and medium strength (1×, 0.1×, 0.01×, 0.001×; TSB and TSA) on transfer of antimicrobial resistance to recipient *Salmonella* was assessed for poultry litter bacteria or *E. coli* MC4100 with pR100-1 as donors as follows. Bacteria were grown in regular-strength TSB, supplemented with MgSO_4_, and incubated overnight at 37 °C. Mating mixes were made as previously described. Filters were placed on regular strength or diluted (10-to-1000 fold) TSB with agar (1.5% wt./volume) (Fisher), supplemented with 10 mM MgSO_4_, and incubated overnight at 37 °C. Matings were also performed placing filters on standard TSA with 10 mM MgSO_4_ and incubating plates at 25 °C or 37 °C overnight; filter matings were performed at 25 °C for 24 and 72 h. Recipients and transconjugants from matings with *E. coli* R100 or litter bacteria were enumerated by plating dilutions on TSA with antibiotic combinations and concentrations as stated previously. All dilutions were plated in triplicate.

In order to determine the minimum *Escherichia* pR100 donor cell density with poultry litter bacteria sufficient to observe plasmid transfer, it was assessed as follows. *Salmonella* recipient and *E. coli* donor strains were grown overnight in TSB with 10 mM MgSO_4_. Recipient *S*. Typhimurium pSLT^−^ strain (10^6^ or 10^8^ CFU/mL) was mixed with *E. coli* R100-1 plasmid donor that had been diluted 10-fold to cell densities ranging from 10^7^–10^0^ CFU per ml in 5 mL 10 mM MgSO_4_ with litter bacteria. The poultry litter bacteria were co-incubated with *S*. Typhimurium pSLT- recipient at 10:1 (10^8^ to 10^6^ CFU/mL) or 1:10 (10^6^ to 10^8^ CFU/mL) ratios of poultry litter bacteria to *Salmonella* and *E. coli* MC4100 with pR100-1 (10^6^–10^0^ CFU/mL). Cells were collected on 0.45 μm filters; filters were aseptically transferred to TSA; and plates were incubated overnight at 37 °C. The mating mix was plated on TSA with rifampicin (64 μg/mL) or rifampicin plus chloramphenicol (25 μg/mL) to enumerate recipients and transconjugants, respectively. All dilutions were plated in triplicate.

### 4.4. DNA Extraction and qPCR

DNA was extracted from 10^8^ bacteria cells using the ZymoBIOMICS DNA Miniprep Kit (Zymo Research, Irvine, CA, USA) according to the manufacturer’s instructions. The length of bead-beating was optimized for DNA extraction from poultry litter bacteria by comparing incremental vortex time (0, 5, 10, 20, 30, or 40 min) with visual assessment of the decrease in bacterial cell density monitored by Gram stain and microscopy. DNA was quantified using a NanoDrop One (Thermo Fisher Scientific; Waltham, MA, USA). There was significant bacterial lysis only after 10 min of vortexing and DNA concentration plateaued at this time point as compared to the later time points. Ten minutes was therefore chosen as the optimal length for bead-beating using this kit. Aliquots of DNA were made in fresh tubes and were normalized to 10 ng/µL using molecular-grade water. DNA samples were stored at −20 °C. To estimate total bacterial genomes, 16S rRNA qPCR was used, and qPCR was used to quantify streptomycin-resistance, sulfonamide-resistance, and class 1 integron-integrase genes *aadA*, *sul1*, and *intI1*, respectively, in the poultry litter community (Table 2). qPCR was performed in triplicate. A standard curve was generated using pDU202 for *aadA1* and *sul1* and *E. coli* LE392 for 16S rRNA, starting with 10 ng DNA template and a series of ten-fold dilutions to 1 picogram. Every PCR experiment included a tube without DNA template (no-template) in order to detect reagent or PCR contamination. *Escherichia coli* strain LE392 served as an additional negative control for the *aadA1* and *sul1* qPCR and, like the no-template control, was never positive. qPCR mixtures contained 5 µL of iQ SYBR Green Supermix (Bio-Rad; Hercules, CA, USA), 0.5 µM of forward and reverse primers, 1 µL of template DNA, and molecular-grade water for a final volume of 10 µL. The following thermocycler (Bio-Rad CFX96 Real Time System; C1000 Touch ^TM^ thermocycler) conditions were used: 3 min at 94 °C and 30 cycles of 30 s at 94 °C, 30 s at annealing temperatures stated in Table 6, and 2 min at 72 °C. This was followed by a melt curve with a temperature range of 55 to 95 °C and 0.5 °C increments. The peak melt curve for all positive litter samples overlapped with the positive controls for *aadA*1 and *sul1*. The melt curve for the 16S rRNA qPCR produced a distinct, non-overlapping peak for the litter samples compared to the *E. coli* LE392 control. The peaks for all the litter samples overlapped. This result was not unexpected as Gram-positive Firmicutes and Actinobacteria are the dominant phyla in poultry litter [14].

### 4.5. Poultry Litter Microcosm

Frozen poultry litter from pooled field samples were placed in Whirl-PAK bags (Nasco; Fort Atkinson, WI, USA) and pulverized with a Seward Stomacher^®^ 400 Circulator stomacher (Seward Ltd.; Norfolk, UK) (10× *g* for 5 min). Fifteen grams was subsequently placed in a 50 mL conical tube. Each tube received 12 mL of 10 mM MgSO_4_ containing *E. coli* plasmid donor (10^7^ or 10^4^ CFU/mL), *S*. Typhimurium pSLT^−^ recipient (10^7^ or 10^6^ CFU/mL), donor and recipient, or no bacteria added. Each tube was then vortexed for 1 min to ensure even distribution liquid into poultry litter. Samples were then incubated for a total of 7 or 14 days with 1 g sampling, performed in triplicate for each sample at days 0, 1, 3, 7, and 14. Bacterial extractions were preformed from each sample as previously described. The cell suspensions were diluted 10-fold and plated on TSA plates containing chloramphenicol (pR100-1; 25 μg/mL) or kanamycin (pRSA; 50 μg/mL) alone, rifampicin alone (64 μg/mL), or combination chloramphenicol (pR100-1) or kanamycin (pRSA) and rifampicin for enumerating donor, recipient, and transconjugants, respectively. All dilutions were plated in triplicate.

### 4.6. Statistical Analysis

Analysis of Variance (ANOVA) was used to determine differences between and within groups. Student’s *t*-test was also used to determine significant differences between groups. Linear regression was used to determine correlation between transconjugant abundance and recipient, donor abundance, or media strength.

## 5. Conclusions

Transmittance of AMR within any microbial community is dependent on several factors. Key to this is that the resistance needs to reside on conjugative, stable MGE with a broad host range and a competent donor host capable of persisting in this environment. However, conjugation is just one mechanism by which AMR spreads among microbial communities. Phages and natural transformation also play an important role in the spread of AMR [92,93].

The reality is distribution of antimicrobial resistances and their associated resistance genes is uneven across the diverse microbial species that inhabit poultry litter [94]. While specific tetracycline resistance alleles such as *tetM* are found among phylogenetically diverse phyla in animal manures [95,96], other alleles have co-evolved with their bacterial host: *tetQ* in the *Bacteroidia* [95] or the tetracycline efflux pumps and associated *tet* alleles *tetA*, *tetB*, or *tetC* in the *Enterobacteriaceae* [96]. AMR do find their way into pathogens such as *S. enterica*, often on conjugative plasmids that vary in host range from the narrow incF to broad-host-range incQ conjugative plasmids [97]. Enteropathogens, such as *Salmonella*, do acquire AMR from community bacteria whether it is environmental or gut microbiota. The challenge is to prevent emergent resistance or spread of AMR pathogens in the environment. The key might be litter management for pathogen-exclusion properties [98,99] or processes, such as composting, that reduce pathogens in poultry litter [100,101].

## Figures and Tables

**Figure 1 antibiotics-12-01006-f001:**
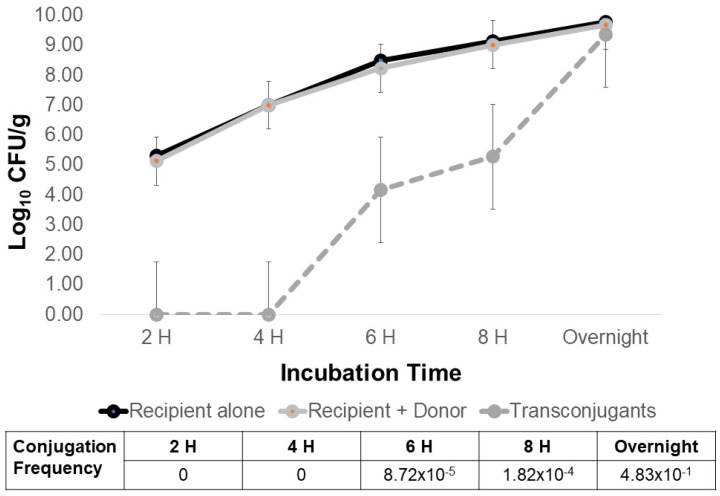
Optimal plasmid transfer incubation time for incFII plasmid R100-1 at 37 °C in filter matings with rifampicin-resistant *Salmonella* as a recipient. Filter matings were performed between the plasmid donor, nalidixic-acid-resistant *E. coli* MC4100 containing R100, and the *Salmonella* recipient *S*. Typhimurium LT2 pSLT^−^ at a 1:10 ratio. *Salmonella* recipient and transconjugants were enumerated by plating 10-fold dilutions onto rifampicin (64 μg/mL) alone or with chloramphenicol (25 μg/mL). Conjugation frequency was calculated as #transconjugants/#recipients.

**Figure 2 antibiotics-12-01006-f002:**
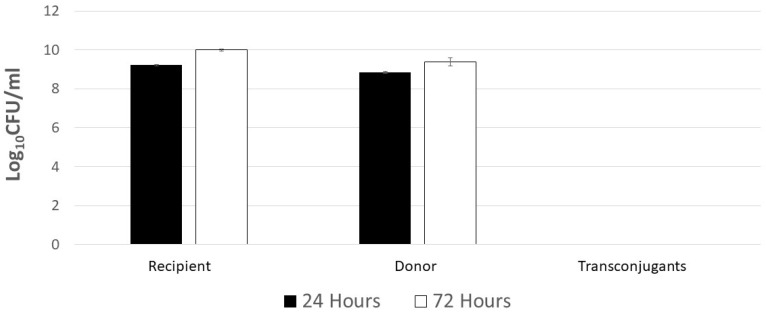
Plasmid R100 does not transfer at 25 °C even with lengthened 72 h incubation. Rifampicin-resistant *S.* Typhimurium LT2 pST^−^ was the recipient in filter matings with donor nalidixic acid-resistant *E. coli* MC4100 with R100. *Salmonella* and donor bacteria were mixed at 1:10 ratio. Mating mixes were filtered with 0.45 μm filters and placed on tryptic soy agar (TSA); where plates were incubated at room temperature for 24 or 72 h. Donors and recipients were enumerated on TSA plates containing rifampicin (64 μg/mL) or chloramphenicol (25 μg/mL), respectively. Transconjugants were enumerated from TSA containing chloramphenicol and rifampicin. No transconjugants were observed.

**Figure 3 antibiotics-12-01006-f003:**
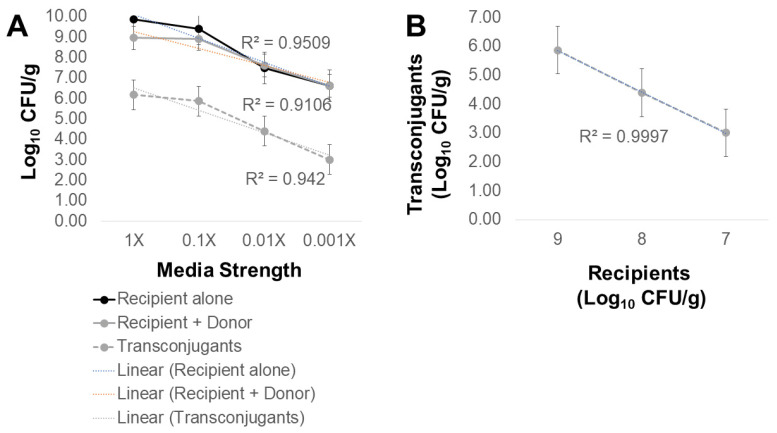
(**A**) Effect of medium concentration on transfer of R100-mediated antimicrobial resistance to *Salmonella* at 37 °C. Tryptic soy agar (TSA) was used as per manufacturer’s recommendations (1×) or diluted 10- (0.1×), 100- (0.01×), or 1000-fold (0.001×); to this, MgSO_4_ (10 mM) and agar (1.5%) was added. Rifampicin-resistant *S*. Typhimurium LT2 pSLT^−^ and nalidixic-acid-resistant *E. coli* containing R100 or poultry litter bacteria, in a 1:10 recipient to donor ratio, were filtered onto 0.45 μm filers. Poultry litter bacteria were isolated from litter collected from the 3rd successive research chicken flock raised on built-up poultry litter. Conjugation frequency of R100 to *Salmonella* was calculated as #transconjugants/#recipients. Recipients and transconjugants were enumerated by plating 10-fold dilutions onto media with rifampicin (64 μg/mL) alone or with ampicillin (25 μg/mL), chloramphenicol (25 μg/mL), streptomycin (100 μg/mL), or tetracycline (25 μg/mL). No transconjugants were observed for any of the antibiotic combinations in filter matings with *Salmonella* and poultry litter bacteria. (**B**) Linear correlation between transconjugants and recipient abundance in filter matings.

**Figure 4 antibiotics-12-01006-f004:**
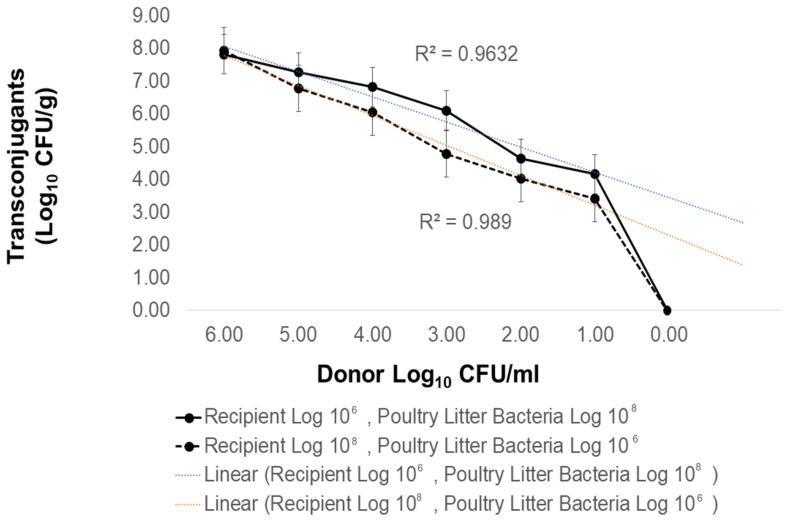
Plasmid R100 transfer to rifampicin-resistant *Salmonella* in the presence of poultry litter bacteria. Bacteria were extracted from poultry litter collected from 3rd successive research chicken flock raised on built-up litter. Recipient *Salmonella* LT2 pSLT^−^ at 10^6^ (solid line) or 10^8^ (dotted line) were mixed with poultry litter bacteria at 10^8^ or 10^6^ and donor nalidixic-acid-resistant *E. coli* MC4100 R100 at varying cell densities (X-axis) on a 0.45 μm filter. Filters were aseptically transferred onto TSB agar with 10 mM MgSO_4_ and incubated for 24 h at 37 °C. Cell suspensions were diluted 10-fold and plated on rifampicin alone (64 μg/mL) or with chloramphenicol (25 μg/mL).

**Figure 5 antibiotics-12-01006-f005:**
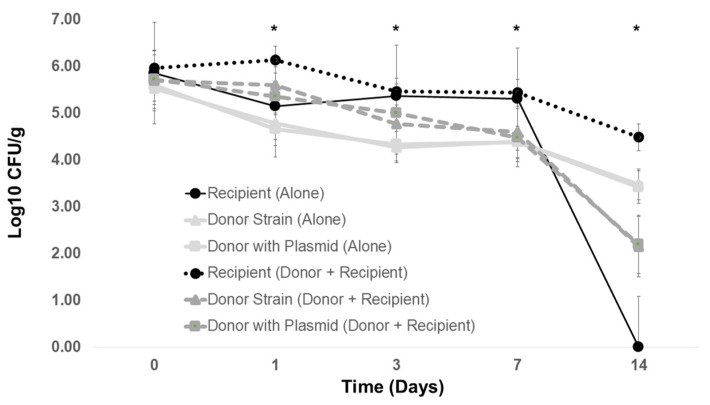
Transfer of plasmid R100 to recipient *Salmonella* in poultry litter at 25 °C. Rifampicin-resistant *S.* Typhimurium LT2 pSLT^−^ was added to poultry litter alone or with plasmid donor *Escherichia coli* R100 (*incF*). Poultry litter was obtained from four commercial broiler farms and pooled. The donors and recipients were enumerated by plating 10-fold dilutions onto tryptic soy agar (TSA) with chloramphenicol (25 μg/mL) or rifampicin (64 μg/mL), respectively. Transconjugants were enumerated from TSA containing both chloramphenicol and rifampicin. No *Salmonella* transconjugants were observed from poultry litter bacteria or from the pR100 donor. * Group differences as determined by ANOVA, *p* < 0.01.

**Figure 6 antibiotics-12-01006-f006:**
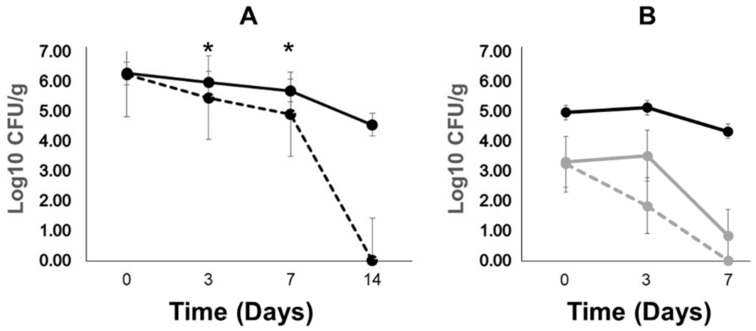
Different *E. coli* donor strains’ persistence in poultry litter microcosm at 25 °C. *E. coli* donor 1932 or poultry isolate 5651, both nalidixic-acid-resistant, served as plasmid pRSA (kanamycin resistance) donors to recipient rifampicin-resistant *S.* Typhimurium LT2 pST^−^. The donors and recipients were enumerated by plating 10-fold dilutions onto tryptic soy agar (TSA) with kanamycin (50 μg/mL) and nalidixic acid (64 μg/mL) or rifampicin (64 μg/mL). No transconjugants were detected on TSA containing both kanamycin (50 μg/mL) and rifampicin (64 μg/mL). (**A**) donor *E. coli* 1932 containing plasmid pRSA. Closed circles represent recipient rifampicin-resistant *S.* Typhimurium LT2 pST^−^. Dotted line: recipient *Salmonella* alone; solid line: donor mixed with recipient. No transconjugants were detected. *Escherichia coli* 1932 with pRSA was detected in poultry litter at 4.90 log10 CFU/g at day 0, but was not detected at later time points. * Student’s *t*-test *p* < 0.01. (**B**) *E. coli* poultry isolate 5651 served as plasmid pRSA donor. Black, closed circle with solid line: *Salmonella* LT2 pST^−^; gray, closed circle with solid line: nalidixic-acid-resistant *E. coli*; and gray, closed circle with dashed line: kanamycin-resistant *E. coli* (pRSA). There were no significant differences (Student’s *t*-test, *p* > 0.05) in *E. coli* counts on nalidixic acid or kanamycin. No transconjugants were detected. As low to no *E. coli* were detected in poultry litter at 7 days, the experiment was terminated after this time point. Poultry litter was obtained from four commercial broiler farms and pooled.

**Table 1 antibiotics-12-01006-t001:** Antimicrobial resistance (AMR) of poultry litter bacteria and abundance of class 1 integron-associated resistance genes *aadA1* and *sul1*.

Sample ^a^	Total Aerobic Counts (Log10 CFU/g) ^b^	Gram-Negative Bacteria (Log_10_ CFU/g) ^c^	Antimicrobial Resistance (Log_10_ CFU/g) ^d^				AMR Gene Abundance ^e^	
			Cm	Ap	Sm	Tc	*aadA1*/BG ^g^	*sul1*/BG ^g^
1	8.56 + 7.58	<2.99 ^f^	4.46 ± 3.47	<2.99	6.20 ± 5.36	7.30 ± 6.48	−2.37 ± −3.29	−0.46 ± −1.37
(Range)	8.00–8.90 ^h^	-	4.00–4.78 ^i^	-	5.90–6.60 ^h^	6.73–7.70 ^j^	−2.22–2.62 ^i^	−0.92–0.61 ^h^
2	8.81 ± 8.10	4.63 ± 4.12	-	7.57 ± 7.29	7.19 ± 6.93	7.25 ± 6.60	−3.42 ± −3.95	−1.15 ± −2.20
(Range)	8.15–9.11 ^j^	3.00–4.90 ^j^	<2.99–3.78	4.00–8.30 ^j^	4.60–8.30 ^j^	5.50–7.78 ^j^	−4.00–2.91 ^j^	−1.45–0.91 ^i^
*t*-test, *p* = ^k^	0.0200	ND	ND	ND	0.0604	0.3196	1.22 × 10^−5^	3.43 × 10^−5^

^a^ Pooled poultry litter samples from (1) 3rd successive chicken research flock raised on built-up litter (*n* = 3); (2) built-up litter of commercial broiler chicken farms (*n* = 4). ^b^ Colony counts on tryptic soy agar, grown overnight at 37 °C. ^c^ Total colony counts on MacConkey agar incubated overnight at 37 °C. ^d^ Cm—chloramphenicol (25 μg/mL); Ap—ampicillin (25 μg/mL); Sm—streptomycin resistance (100 μg/mL); and Tc—tetracycline (25 μg/mL). ^e^ Copy # determined from standardized curve using *Escherichia coli* MC4100 containing plasmid R100 as the standard. DNA concentration was normalized to 30 ng in qPCR. AMR gene abundance was presented as Log_10_ ratio of AMR gene copies to 4.2 16S genes per bacterial genome. [43]. ^f^ Limit of detection. ^g^ BG—bacterial genomes. ^h^ ANOVA *p* > 0.05. ^i^ ANOVA *p* < 0.05. ^j^ ANOVA *p* < 0.01. ^k^ Comparison of poultry litter from commercial vs. experimental sources.

**Table 2 antibiotics-12-01006-t002:** Transfer of antimicrobial resistance from poultry litter bacteria to *Salmonella enterica* Typhimurium LT2 in vitro.

Mating Mix Combination	Recipients (Log10)	Transconjugants (Log10) ^a^			Conjugation Frequency ^b^
		Cm ^c^	Sm ^c^	Tc ^c^	
*Salmonella* LT2 (recipient control) ^d^	9.14 + 8.14	<2.99 ^e^	<2.99 ^e^	<2.99 ^e^	0.0
*Escherichia coli* R100 (donor control)	<2.99 ^e^	<2.99 ^e^	<2.99 ^e^	<2.99 ^e^	0.0
*Salmonella* LT2 + *E. coli* R100 ^f^	9.03 ± 8.25	7.56 ± 6.70	7.61 ± 6.77	7.61 ± 6.72	3.32 × 10^−2^
*Salmonella* LT2 + Litter 1 ^g^	9.16 ± 6.24	<2.99 ^e^	<2.99 ^e^	<2.99 ^e^	0.0
*Salmonella* LT2 + Litter 1 ^g^ + *E. coli* R100 ^f^	9.30 ± 8.39	7.61 ± 7.14	7.46 ± 7.06	7.44 ± 6.61	1.2 × 10^−2^
*Salmonella* LT2 + Litter 2 ^g^	7.42 ± 5.59 ^h^	<2.99 ^e^	<2.99 ^e^	<2.99 ^e^	0.0
*Salmonella* LT2 + Litter 2 ^g^ + *E. coli* R100 ^f^	7.41 + 6.67 ^h^	3.43 + 2.71 ^h^	3.43 + 2.71 ^h^	3.43 + 2.95 ^h^	9.30 × 10^−5^

^a^ antibiotic x + rifampicin (64 μg/mL). ^b^ #transconjugants/#recipients. ^c^ Cm—chloramphenicol (25 μg/mL); Sm—streptomycin resistance (100 μg/mL); and Tc—tetracycline (25 μg/mL). ^d^ Rifampicin-resistant (64 μg/mL). ^e^ Limit of detection. ^f^ 1:10 donor to recipient ratio. Mating mixes were filtered onto 0.45 μm filters. The filters were transferred to TSA plates containing 10 mM MgSO_4_ and incubated overnight at 37 °C. ^g^ Pooled poultry litter samples collected from (1) 3rd successive research chicken flock raised on built-up poultry litter; (2) built-up litter of commercial broiler chicken farms. ^h^ Comparison of poultry litter from commercial vs experimental sources; Student’s *t*-test, *p* < 0.01.

**Table 3 antibiotics-12-01006-t003:** Plasmid transfer rate resulting from varying mating conditions.

Mating Mix Combination	Conjugation Frequency ^a^			
	25 °C		37 °C	
	2 H	24 H	2 H	24 H
*Salmonella* LT2 (recipient alone)	0.00	0.00	0.00	0.00
*Escherichia coli* pR100 (donor alone)	0.00	0.00	0.00	0.00
*Salmonella* LT2 + *E. coli* pR100	0.00	0.00	0.00	1.43 × 10^−1^
*Salmonella* LT2 + Litter 1 ^b^	0.00	0.00	0.00	0.00

^a^ #transconjugants/#recipients. Recipients and transconjugants were enumerated by plating 10-fold dilutions onto media with rifampicin (64 μg/mL) alone or combination chloramphenicol (25 μg/mL) and rifampicin (64 μg/mL), respectively. ^b^ Poultry litter collected from 3rd successive research chicken flock, raised on built-up litter.

**Table 4 antibiotics-12-01006-t004:** Plasmid transfer frequencies are dependent on plasmid type, donor strain, and temperature.

*Escherichia coli* Strain	Plasmid ^3^	Temperature	
		25 °C ^4^	37 °C ^4^
MC4100	pR100	0.00	1.43 × 10^−1^
1932 ^1^	pRSA	1.36 × 10^−4^	4.90 × 10^−3^
5651 ^2^	pRSA	3.59 × 10^−5^	ND
8762 ^2^	pRSA	0.00	ND
9270 ^2^	pRSA	0.00	ND
1932 ^1^	pSDb1	8.05 × 10^−4^	1.73 × 10^−7^
1932 ^1^	pSDb2	0.00	0.00
1932 ^1^	pSHb1	2.10 × 10^−5^	2.32 × 10^−6^
1932 ^1^	pSKy1	5.97 × 10^−9^	2.23 × 10^−8^
1932 ^1^	pSKy2	2.90 × 10^−9^	1.04 × 10^−4^
1932 ^1^	pSNp1	0.00	1.69 × 10^−4^
1932 ^1^	pSTm1	4.73 × 10^−4^	2.24 × 10^−6^

^1^ Human isolate [52]; ^2^ chicken isolate [51], ^3^ pR100 (chloramphenicol resistance); pRSA (kanamycin resistance) plasmid; pSDb1–pSTm1 (tetracycline resistance) plasmids. ^4^ #transconjugants/#recipients. *S.* Typhimurium LT2 pST^−^ served as recipient in filter matings that do not include litter bacteria. Donor to recipient ratios of 1:1 were used in filter matings with transconjugants detected on tryptic soy agar (TSA) containing rifampicin (64 μg/mL) with either chloramphenicol (25 μg/mL), kanamycin (50 μg/mL), or tetracycline (25 μg/mL).

**Table 5 antibiotics-12-01006-t005:** Bacterial strains and plasmids.

Bacterial Strain or Plasmid	Description ^a^	References
*Salmonella enterica*		
pSLT^−^	*S.* Typhimurium LT2 strain; Rif^r^, lacking the *spvB*^−^ virulence plasmid	[78]
*Escherichia coli*		
MC4100	Sm^r^; F^−^ *supE44* ∆*lacU169* (ϕ80 l*acZ*∆*M15*) *hsdR17 recA1 endA1 gyrA96 thi01 relA1*; Nal^r^	
1932	Human isolate; Nal^r^	[52]
5651	Chicken isolate; Nal^r^	[51]
Plasmids		
pR00	IncFII plasmid; Cm^r^ Fa^r^ Sm^r^ Sp^r^ Su^r^ Tc^r^, Tra^+^ conjugative	[89]
pDU202	tetracycline-sensitive derivative of R100-1; *intI1*, *aadA1*, *sul1* qPCR control	[90]
pRSA	Broad-host-range, conjugative IncW plasmid encoding Cm^r^, Km^r^	[49]
pSDb1	*S.* Dublin strain CVM 22429; IncA/C, IncI1 plasmid encoding extended spectrum cephasporinase, Cm^r^, Tc^r^, Sm^r^	
pSDb2	*S.* Dublin strain N16S275; IncA/C, IncFII, IncX plasmid encoding extended spectrum β-lactamase, Cm^r^, Sm^r^, Tc^r^	
pSHb1	*S.* Heidelberg strain N16S201; IncA/C, IncI1 plasmid encoding extended spectrum β-lactamase, Sm^r^, Tc^r^	
pSKy1	*S.* Kentucky strain N162104; IncFII, IncF1b, IncI1, IncXI plasmid encoding extended spectrum β-lactamase, Sm^r^, Tc^r^	
pSKy2	*S.* Kentucky strain N173914; incFIB, IncFII, IncI1, IncX1 plasmid encoding extended spectrum cephasporinase, Km^r^, Tc^r^	
pSNp1	*S.* Newport strain N17S1196; IncA/C plasmid encoding extended spectrum β-lactamase, Cm^r^, Sm^r^, Tc^r^	
pSTm1	*S.* Typhimurium strain N17S520 IncI1 plasmid encoding extended spectrum cephasporinase, Sm^r^, Su^r^, Tc^r^	

^a^ Ap, ampicillin; Cm, chloramphenicol; Fa, fusaric acid; Nal, nalidixic acid; Rif rifampicin; Sm, streptomycin; Sp, spectinomycin; Su, sulfonamide; Tc, tetracycline.

**Table 6 antibiotics-12-01006-t006:** PCR primers.

Target Gene	Sequence	Amplicon Size (bp)	Annealing Temp (°C)	Reference
16S	F: CGGTGAATACGTTCYCGG	142	56.3	[91]
	R: GGWTACCTTGTTACGACTT			
*aadA1*	F: GTACGGCTCCGCAGTGGA	244	56.3	[18]
	R: GCGCTGCCATTCTCCAAA			
*sul1*	F: TTGGGGCTTCCGCTATTGGTCT	187	62.0	[18]
	R: GGGTTTCCGAGAAGGTGATTGC			

## Data Availability

Not applicable.

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
