# Peer review of "Conditions Necessary for the Transfer of Antimicrobial Resistance in Poultry Litter"

_antibiotics, 2023, doi:10.3390/antibiotics12061006_

Round 1

Reviewer 1 Report

Although the AMR is reported. The experiments  experiments of conjugation did not permit to afirm that conjugation does occur in poultry litter. What would be  conditions for conjugation to occur? I also would recommend to discuss other possibilities of gene transfer.

Author Response

Reviewer 1

I had issues with track changes in the “Antibiotics” formatted manuscript.  I therefore highlighted the changes in yellow.

Although the AMR is reported, the experiments of conjugation did not affirm that conjugation does occur in poultry litter.

What would be conditions for conjugation to occur?

RESPONSE: “The inability to document plasmid transfer from the poultry litter microbiome to Salmonella may be attributed to: 1) the absence of AMR conjugative plasmids in the bacterial population (73); 2) limited transfer potential of conjugative plasmids or plasmid replication outside their donor host, kin or evolutionarily-related bacteria (37, 68, 74-76); or 4) genetic barriers to plasmid acquisition (77), including exclusion by resident plasmids (78).” Lines 197-201.

“The IncW plasmid pRSA was chosen based on its use in comparable studies (49, 79) and its ability to transfer at 25oC.  However, transfer was not observed when E. coli with conjugative plasmid pRSA was used in poultry litter.  In this case, E. coli strain type became a confounding factor as E. coli 1932 died off after 3 days.  While switching to a poultry isolate, E. coli 1932, resulted in improved persistence, no plasmid transfer was detected.  Others have also reported the influence of donor strain and plasmid type on AMR transfer (68, 69, 74, 80, 81).  While the focus here is on donor and plasmid, recipient background can significantly impact on plasmid acquisition (74, 78) therefore we selected Salmonella LT2 pST- which has been cured of the resident F plasmid that can act as barrier to plasmid transfer.” Lines 208-217.

“Alternatively, the poultry litter microbiota may produce factors that interfere with plasmid abundance or transfer (73).  Bacterial growth is central to AMR transfer (45) as energy is needed for plasmid transmission as well as DNA synthesis machinery needed to produce the complementary strand of the transferred plasmid DNA (82).  Because no increase in donor or recipient cell density was ever observed in poultry litter even at low cell densities, transfer may be a rare event. But even in an E. coli strain that persisted in poultry litter, the plasmid was lost after 3 days of incubation in the microcosm.  We previously reported similar findings for S. Newport AMR plasmid in chickens administered E. coli donors with antibiotic selection pressure (42). Others have reported the impact of donor strain background on plasmid transfer (69, 74, 80, 81).” Lines 221-231.

We also added the following statement in our revision. “Plasmid transfer in litter requires: 1) permissive microbiota, bacteria strain and plasmid types; 2) bacteria growth; 3) water activity that favors biofilm formation; and antibiotic selection pressure that favors emergent AMR in pathogens inhabiting this environment.” Lines 240-245.

I also would recommend to discuss other possibilities of gene transfer.

RESPONSE:  While plasmid transfer in litter was the primary focus of this study, we agree that other mechanisms like transformation and phage transduction have played an important role in the evolution of AMR. We made the following changes.

“Many AMR genes are also linked to mobile genetic elements (MGE) (53, 55) however integrons and some transposons are not self-transmissible and dependent on conjugative plasmids, phages or natural transformation for dissemination.” Lines 160-162.

We also added the following to Conclusions.

“Transmittance of AMR within any microbial community is dependent on several factors.  Key to this is that the resistance needs to reside on conjugative, stable MGE with a broad host range and a competent donor host capable of persisting in this environment.  However, conjugation is just one mechanism by which AMR spreads among microbial communities.  Phages and natural transformation also play an important role in the spread of AMR (92, 93).” 5. Conclusions: Lines 2-8

Reviewer 2 Report

The authors have evaluated the poultry resistome, the transfer of plasmids to Salmonella (bacteria of public health concern) and the experimental conditions to study conjugation experiments. Overall, the study is relevant to antimicrobial resistance research. However, there are major deficiencies that need correction:

1. Introduction is adequate and well-written.

2. Methods: two key paragraphs are missing: A) description of the farms themselves. b) statistical analysis

3. Results: Results are overall well-presented in written. However, there are absolutely zero statistics applied, making the whole study descriptive and not inferential. Are the error bars std. deviation or error? What were the number of replicates? None of this is mentioned, and should be improved.

4. Discussion and conclusions are well written.

Author Response

Reviewer 2

I had issues with track changes in the “Antibiotics” formatted manuscript.  I therefore highlighted the changes in yellow.

The authors have evaluated the poultry resistome, the transfer of plasmids to Salmonella (bacteria of public health concern) and the experimental conditions to study conjugation experiments. Overall, the study is relevant to antimicrobial resistance research. However, there are major deficiencies that need correction:

  1. Introduction is adequate and well-written.
  2. Methods: two key paragraphs are missing: A) description of the farms themselves. b) statistical analysis

RESPONSE:  We wrote the following to better describe the source of poultry litter used in this study. “Poultry litter used in this study was obtained from two sources: experimental broiler flocks and commercial broiler chicken houses in the southeastern United States.  Litter obtained from experimental flock was collected following 3rd successive chicken flock raised on built up litter. Random grabs of poultry litter were collected from vacated experimental pens and pooled.  Poultry litter was also obtained from four commercial broiler chicken farms, submitted to us by a third party.  The chicken breeds were typical Ross or Ross/Cobb hybrids used for the production of meat birds (broilers). Bedding for experimental and commercial flocks was pine shavings.  Commercial broiler production in the US typically uses built-up litter system: wind-rowing to compost litter before its spread, to which it is top dressed with fresh pine shavings before the placement of the next flock.” Lines 265-275.

  1. Results: Results are overall well-presented in written. However, there are absolutely zero statistics applied, making the whole study descriptive and not inferential. Are the error bars std. deviation or error? What were the number of replicates? None of this is mentioned, and should be improved.

RESPONSE: First, I poured through student laboratory notebooks and ExCell spreadsheet and did ANOVA, T-test and linear regression on data sets where appropriate.  In doing so, I caught some mistakes and made changes accordingly.  All dilutions were plated in triplicate.  For the poultry microcosm studies, three 1-gram samples were taken from microcosm at different time points.  All error bars, in figures are standard deviation.  This has been added to Materials and Methods and highlighted in yellow, and highlighted as well in figures, tables and results sections.

  1. Discussion and conclusions are well written.

Round 2

Reviewer 2 Report

Thank you for incorporating the suggested changes.